# Employing DEA for Assessment of Cruise Market: A Case Study in Malaga—Spanish Port

**José Ignacio Parra Santiago** [1,*]📍, **Alberto Camarero Orive** [2]📍, **David Díaz Gutiérrez** [1] **and Francisco de Asís De Manuel López** [1]

1   Escuela Técnica Superior de Ingenieros Navales, Universidad Politécnica de Madrid, 28040 Madrid, Spain
2   Escuela Técnica Superior de Ingenieros de Caminos, Canales y Puertos, Universidad Politécnica de Madrid, 28040 Madrid, Spain
*   Correspondence: joseignacio.parra.santiago@upm.es

**Abstract:** In recent decades, the Spanish port system has been investing in the development of infrastructures aimed at attracting the cruise market. For this reason, this paper analyses, using a methodology based on data envelopment analysis (DEA), the efficiency of the Spanish port system for the cruise market. Most of the port authorities want to attract and maintain this traffic in their ports, due to the economic impact it has on them. Based on the data provided by Puertos del Estado and the port authorities in their annual reports, such as the number of cruise passengers per port or port authority, the number of stopovers or ships that visit our ports each year, and the infrastructures that have been developed for this market, an efficiency analysis was carried out to draw conclusions at the level of the port system and each port authority in terms of operational efficiency. Focusing the research on Malaga, the current situation was analyzed, as well as the forecasts that resulted from the research for the future development of the port in terms of cruise ships.

**Keywords:** port system; efficiency; DEA-bootstrapping; maritime traffic; Spain; cruise

## 1. Introduction

Spain, one of the world's main tourist destinations, is the focus of the cruise market due to its geographical position in the Western Mediterranean, the Atlantic coast, and the Canary Islands. This tourist attraction is the main engine of the Spanish economy, and for this reason, our port system is aware of and invests in the necessary infrastructures to attract cruise ships and a greater number of tourists to our territory. The impact of tourism represents around 12% of the country's gross domestic product (INE, National Statistics Institute), with part of its economy being sustained by the service sector, with a development and excellence unparalleled in the hotel and transport offers.

The cruise market is a market that is very sensitive to political and social events in the territories they visit, as the instability that this causes can make passengers choose one route or another, which is reflected in economic income, with the "Arab Spring" standing out as a political event that modified a large number of cruise routes [1] (Carvajal Pineda 2015). It is also very sensitive to accidents that may happen, as the media and social repercussions affect the demand for hiring these holiday services, as happened with the Costa Concordia accident [2] (Schröder-Hinrichs, Hollnagel, and Baldauf 2012). Therefore, after the COVID-19 crisis, the recovery of the sector will be gradual, as most people will opt for domestic tourism that does not involve the confinement of large numbers of people (as in the case of cruises).

For years, main shipping lines and cruise liners have been choosing Spanish ports as stopovers on their routes or, in some cases, as the start/end port of a route (base port). Barcelona is the main Mediterranean cruise port and one of the great European and world ports, as it is chosen for its infrastructures and geographical position, among other aspects,

to start or end a route around the Western Mediterranean, including great routes such as the Spanish-American route, which, from Barcelona, takes the cruise ship to the Americas (Argentina, the Caribbean, or the United States), or other routes of shorter duration but of great importance, such as the Greek islands or the Turkish coast or the Balkans.

This paper analyses, from an empirical point of view, port efficiency in terms of the number of cruise ships that can be accommodated in the set of infrastructures designed for this traffic, and in terms of the number of cruise passengers that they receive throughout their ports [3] (Camarero et al. 2022).

To this end, the efficiency of both the Spanish port system and of a selected group of port authorities that make it up were analyzed in order to obtain a significant sample of ports that have been investing larger amounts of their port budgets in improvements or extensions of their infrastructures for the cruise market. Consequently, most of the ports in this research are placed on the Mediterranean coast, as this is where one of the main cruise tourist destinations in Europe, the Western Mediterranean, is placed.

In addition, the research presents the "case of Malaga", which details the trajectory of the port in the cruise market, since, in recent years, this port has decided to invest in the creation and adaptation of infrastructures aimed at this traffic, which has encouraged its progressive evolution from a stopover port to its classification as a base port.

The research concludes with one of its main conclusions, which is that the efficiencies obtained show a diversified port system, not centered on cruise traffic in all its ports, which favors the dispersion of cruise ships along the entire coastline and the number of cruise passengers who choose its base ports.

In the case of Malaga, its efficiency is remarkable, which is a sign that they are doing things well, due to the port's bet on being a great base port in the Southern Mediterranean.

## 2. Literature Review

Several studies analyzed the economic importance of cruise tourism and cruise calls (Table 1). Major operational research topics include the optimal cruise route, the cruise port selection process, and the optimal pricing policy of cruise passenger cabins. The supply of services and the locational qualities of cruise ports have also been the subject of attention in the literature.

**Table 1.** Research in the field of cruise ships.

| Year | Author | Scope of the Study |
|------|--------|--------------------|
| 1989 | Hersh, M. and Ladany, S.P. [4] | Optimal scheduling of ocean cruises |
| 1990 | Marti, B.E. [5] | Geography and the cruise ship port selection process |
| 1991 | Ladany, S.P. and Arbel, A. [6] | Optimal cruise liner passenger cabin pricing policy |
| 1996 | Dwyer, L. and Forsyth, P. [7] | Economic impacts of cruise tourism in Australia |
| 1998 | Dwyer, L. and Forsyth, P. [8] | Economic significance of cruise tourism |
| 1998 | McCalla, R.J. [9] | An investigation into site and situation: cruise ship ports |
| 2004 | Dwyer, L., Douglas, N., and Livaic, Z. [10] | Estimating the economic contribution of a cruise ship visit |
| 2004 | Douglas, N. and Douglas, N. [11] | Cruise ship passenger spending patterns in Pacific island ports |
| 2010 | Vaggelas, G.K. and Pallis, A. [12] | Passenger ports: service provision and their benefits |
| 2011 | Gui, L. and Paolo Russoz, A. [13] | Cruise ports: a strategic nexus between regions and global lines—evidence from the Mediterranean |

Source: own elaboration.

The cruise sector offers a particular point of view to understand the evolution of world tourism and represents a symbol of the "massive industrialization"of entertainment activities. There is no doubt that this market is highly capital-intensive and characterized by very high fixed costs for operators, who are looking for high-volume and repeat bookings to fill their capacity [14] (Stabler, Papatheodorou and Sinclair 2010). The market is also characterized by aggressive acquisitions, globalization strategies, and corporate concentration.

The paucity of research on cruise tourism in the academic literature seems unjustified in light of the fact that, although cruise tourism worldwide now accounts for only about 2% of global tourism, the industry has grown faster than many other segments in the last two decades, and its impact on many maritime destinations and port cities is becoming significant. Indeed, many islands (especially in the Caribbean) already receive far more cruise tourists than port stopovers [15] (Wood 2000), and in many other regions, the cruise market is beginning to develop at a fast pace. The process of globalization, in which "geographical restrictions on social and cultural arrangements are fading away" [16] (Waters 1995), seems particularly evident in cruise activities. In this way, cruising in the Caribbean represents an exemplary case: cruise operators in this area are not Caribbean at all, cruise destinations are increasingly controlled by foreign interests, their activity falls outside the jurisdiction of Caribbean states, and the labor force is overwhelmingly non-local. In addition, due to the wide variety of services available on board and the limited time spent in each stopover port, the net benefits of this business for local populations and governments are increasingly criticized [17] (Klein and Roberts 2003).

Research on port efficiency and productivity can be classified into three distinct groups: that which focuses on partial productivity indicators, that which focuses on the research of technological frontiers, and that which focuses on the specialization of a given port in a particular field [18] (González and Trujillo 2005). If we look exclusively at the concept of port efficiency, it is understood that it is linked to productivity, but "detaching" from this concept, the statistical theory used in the measurement of efficiency employs two predominantly computational methods, SFA (stochastic frontier analysis) and DEA (data envelopment analysis).

Charłampowicz and Mańkowski [19] (Charłampowicz and Mańkowski 2020) developed a system of evaluating the economic efficiency of maritime container terminals, in which the following methodological approach was proposed. A conceptual model of the economic efficiency evaluation system of maritime container terminals was proposed as a holistic structure in the sense that it is a subsystem of a higher-level management system (first level), and simultaneously proposal as a system consisting of subsystems or modules (second level), which include subsystems or submodules (third level). The above mentions the structure of the system at the second and third levels, i.e., modules and submodules.

In recent years, many authors have researched the efficiency of port areas [20] (González-Cancelas and Camarero 2009), more specifically in the field of container traffic operations. These works mainly used two methods to measure efficiency: data envelopment analysis (DEA) and DEA-bootstrap models.

Wen et al. [21] (2014), Pham et al. [22] (2016), and Lio and Liu [23] (2018), among others, introduced all uncertain DEA models based on the basic DEA model and uncertainty theory, but applied them to different basic DEA models. However, the studies by Wen et al. [21] (2014) and Lio and Liu [23] (2018) only paused on a set of hypothetical samples, while the research by Pham et al. [22] (2016) was tested and compared with the results obtained from the basic DEA software, as well as directly applied to evaluate the efficiency of the world's main container ports in 2016. Therefore, the paper by Pham et al. [24] (2020) used the DEA uncertainty model of Pham et al. [22] (2016), and it was applied in comprehensive multidimensional research to analyze and evaluate many aspects of the performance of major container ports in the last five years, accompanied by an analysis of excesses in input use or shortages in output production to provide some suggestions that could increase the efficiency of container ports.



González and Trujillo [18] (2005) showed that efficiency and port size had an inverse relationship, using the DEA-BCC model in a sample of nine Spanish ports between 1992 and 2000. González and Trujillo [18] (2005) quantified the evolution of technical efficiency in the provision of port infrastructure services in the main Spanish port authorities dedicated to container traffic. The results showed that the reforms brought about significant improvements in technological change, but that technical efficiency changed little on average. More recently, Gil-Ropero et al. [25] (2019) analyzed efficiency changes in the main container ports of the Iberian Peninsula during the period of 2008–2014, specifically in thirteen Spanish and three Portuguese container ports. Productivity evolution was measured by the Malmquist productivity index using DEA methodology. It was concluded that the increase in container traffic was directly related to an increase in the Malmquist productivity index, but with the influence of other variables, which acted as inputs.

Other research focuses on the assessment of supplier performance in terms of attributes of the triple bottom line of sustainability (economic, environmental, and social) and attributes of COVID-19 pandemic response strategies in their supply chain activities. Thus, it can be concluded that the selection of a potentially sustainable supplier is a complex multicriteria decision-making (MCDM) problem, where MCDM techniques are necessary to narrow down the preliminary set of suppliers to the final choices [26] (Schramm, et al., 2020). Furthermore, in applications and in many real-world circumstances, uncertainty is an unavoidable aspect due to the imprecision of human judgments and the imprecise nature of information. Imprecise sources include unquantifiable, incomplete, and inaccessible data, as well as partial ignorance and experts who may be unwilling or unable to give precise numerical values to comparison judgments [27] (Dang, et al., 2022).

## 3. Methodology

It was the authors Roll and Hayuth [28] (1993) who, for the first time, used the DEA method to analyze port efficiency. They compared data from 20 ports and found that DEA analysis overcomes "certain barriers to the specification of efficiency", although they also pointed out the weaknesses of the system and urged future researchers to resolve them.

This analysis technique, which belongs to the so-called "frontier methods" and is based on linear programming, aims to assess the efficiency and productivity of a set of institutions or individuals, generally called "units" (data management units, DMU). A description of this methodology with radial and non-radial measures can be found, among other works, in Caballero, Gómez, and Sala [29] (2009), Soler i Marco, Hernández Sancho, and Sala Garrido [30] (2009), as well as in the books by Coelli et al. [31] (2005) and Cooper, Seiford, and Tone [32] (2007).

The formulation of the production-oriented DEA analysis used in this article is as follows (Equations (1)–(5)), corresponding to the graphical representation in Figure 1:

$$max\theta + \varepsilon(\sum_{i=1}^{m} S_i^- + \sum_{r=1}^{S} S_r^+) \tag{1}$$

$$\sum_{j=1}^{n}(\lambda_j x_{ij}) + S_i^- = x_{i0} \qquad i = 1, 2, \ldots, m \tag{2}$$

$$\sum_{j=1}^{n}(\lambda_j x_{rj}) - S_r^+ = \theta y_{r0} \qquad r = 1, 2, \ldots, s \tag{3}$$

$$\lambda \geq 0 \qquad with: j = 1, 2, \ldots, n \tag{4}$$

$$\sum_{j=1}^{n} \lambda_i = 1 \tag{5}$$

with:

$y_{r0}$ and $x_{i0}$: the $r_{th}$ output and $i_{th}$ input for a DMUo under evaluation.

$\lambda_j$: the decision variables representing the weights that DMUj would place on DMUo to construct its efficient reference set.

$\theta$: the proportional distance between inputs to the envelope, and thus, the measurement of the technical efficiency index.

$\varepsilon$: the smallest real positive number.

$S_i$ and $S_r$: the possible mismatches or excess factors for each input.

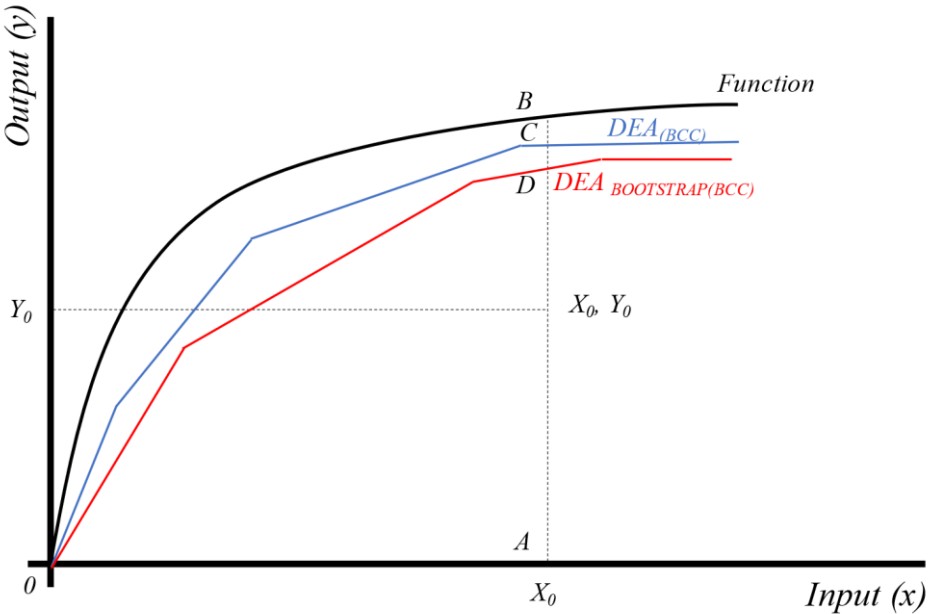

**Figure 1.** Graphical representation of the DEA-bootstrapping model output orientation. Source: based on (Gil Ropero, Turias Dominguez, and Cerbán Jiménez 2019 [25]).

According to López Bermúdez [33] (2018), the mathematical model used in the DEA methodology assumes the existence of n DMUs, each of which consumes m inputs to generate s outputs. Thus, a *DMUj* uses a set of $X_j = x_{ij}$ inputs ($i = 1, \ldots, m$) and generates $Y_j = Y_{kj}$ outputs ($k = 1, \ldots, s$). The s-n matrix of the mean of the output is denoted by $Y$, and the m-n matrix of the inputs is denoted by $X$. It must also be satisfied that $x_{ijj} > 0$ and that $y_{ijj} > 0$.

The method, as previously mentioned, can be output- or input-oriented, thus inverting the nomenclature of the product matrices.

There are usually two types of formulations used in the DEA method, with constant returns to scale (CCR) or with variable returns to scale (BCC). The linear programming approaches of both methods are outlined below:

Using the CCR formulation, one seeks to maximize (Equations (6) and (7)):

$$h_{j0} = \sum_{r=1}^{S} u_r \cdot y_{rj0} / \sum_{i=1}^{m} v_i x_{ij0} \tag{6}$$

subject to:

$$\sum_{r=1}^{S} u_r \cdot y_{rj0} / \sum_{i=1}^{m} v_i x_{ij0} \leq 1, \text{ j } = 1, \ldots, \text{n} \tag{7}$$

$u_r, v_i > 0$, to $r = 1, \ldots, s$ and $i = 1, \ldots, m$

where:

$y_{rj}$ is the quantity of output *r* of unit *j*;

$x_{ij}$ is the quantity of input *i* of unit *j*;

$u_r$ is the output weighting *r*;

$v_i$ is the input weighting $i$;

$n$ is the total number of units;

$s$ is the total number of outputs, and $m$ is the total number of inputs.

The DEA-BOOT efficiency is the result of correcting the value obtained in the simple DEA analysis. With this correction, we bring the value closer to a more accurate approximation, which is more correct with the frontier function, as seen in Figure 1 above.

In this research, the authors tried to demonstrate the efficiency of the investment in infrastructures destined for the cruise market, with the increase in the number of cruise ships visiting their port. This tends to be reflected in about 3–4 years, as the building of infrastructure takes time, and the adaptation to the market derived from this new structure takes time to consolidate. Therefore, this research looked for the novelty of being able to efficiently evaluate some of the main infrastructure actions that the Spanish port system has been carrying out in recent years (namely in 2015 and 2019, because they have been the main years of the cruise market in Spain and in Europe).

### 3.1. Definition of Inputs/Outputs

For this analysis, one input and two outputs were used. The input was related to the infrastructural dimensions, in the years 2015 and 2019, that the different port authorities studied had available to take in and provide service to cruise ships. The outputs used were aimed at attracting this type of traffic. In other words, the total number of cruise passengers that each studied port authority accommodated between 2015 and 2019 was studied, as well as the total number of cruise ships that called at their different ports, also during the same period of years of the study.

For this purpose, they were analyzed using a DEA-bootstrapping model [34] (Simar and Wilson 1998) [35] (Parra et al., 2020) with an output orientation.

The outputs were chosen for their degree of representativeness, as they are data that measure very well the whole of a system or, in greater detail, its components. For this reason, the number of cruise passengers in each port authority and the number of ships received during the year studied are outputs which, in addition to being present in numerous studies, reflect the direct relationship with the linear meters of infrastructure that the different ports use to house these large "floating hotel-cities".

### 3.2. Database Generation

In this section, we attach the table with all the data of the different inputs and outputs extracted from the database of the different port authorities (Table 2), as well as from the database of Puerto del Estado.

**Table 2.** Port data set to determine efficiency.

| Nº | Port Authority | m Dock Year | | Nº Cruisers Year | | Nº Cruises Year | |
|---|---|---|---|---|---|---|---|
| | | 2015 | 2019 | 2015 | 2019 | 2015 | 2019 |
| 1 | Barcelona | 3291.50 | 3291.50 | 2,540,302 | 3,142,664 | 750 | 800 |
| 2 | Tarragona | 707.00 | 707.00 | 12,277 | 128,089 | 8 | 63 |
| 3 | Castellón | 350.00 | 350.00 | 366 | 5462 | 2 | 5 |
| 4 | Valencia | 1654.00 | 1644.00 | 371,374 | 435,616 | 172 | 203 |
| 5 | Baleares | 6618.37 | 5616.72 | 1,958,848 | 2,656,443 | 788 | 818 |
| 6 | Alicante | 957.84 | 957.84 | 82,316 | 63,088 | 50 | 43 |
| 7 | Cartagena | 522.00 | 709.00 | 150,795 | 250,058 | 108 | 167 |
| 8 | Almería | 219.00 | 670.00 | 17,304 | 7177 | 27 | 25 |

**Table 2.** *Cont.*

| Nº | Port Authority | m Dock Year | | Nº Cruisers Year | | Nº Cruises Year | |
|---|---|---|---|---|---|---|---|
| | | 2015 | 2019 | 2015 | 2019 | 2015 | 2019 |
| 9 | Motril | 859.00 | 860.00 | 6481 | 9415 | 27 | 32 |
| 10 | Málaga | 2045.00 | 2045.00 | 418,503 | 476,973 | 238 | 288 |
| 11 | Bahía de Cádiz | 324.00 | 800.00 | 411,455 | 477,387 | 295 | 333 |
| 12 | Huelva | 1009.00 | 1104.00 | 3730 | 1357 | 6 | 7 |
| | Total | 18,556.71 | 18,755.06 | 5,973,751 | 7,653,729 | 2471 | 2784 |

Source: own elaboration with data extracted from the different port authorities, as well as from Puertos del Estado (2015 and 2019).

## 4. Results

The results obtained through the inputs and outputs selected according to their nature allowed us to obtain an overall view of the functioning of our port system, by means of the individual analysis of each port authority studied.

This paper shows the efficiency in terms of cruise ships, for which the berthing and mooring infrastructures were taken into account, in order to measure the efficiency in this very important aspect, which marks the daily operations of the port.

This is unlike the benchmark outputs in a port system in terms of cruises, which are the total number of cruise passengers that the port authority receives in a year, and in this case, the number of cruises was chosen as a representative output of efficiency, as it is directly related to the linear meters of berth used to receive this type of traffic, given that, as the years go by, the ships tend to be larger and deeper draught. This does not mean, however, that the greater the number of ships, the greater the efficiency, but rather that there is a relationship between the number of berth meters and the number of ships that can be accommodated by each port, all depending on the technical characteristics.

With an alpha error equal to 0.05 (5%), the following results are obtained in the efficiency result tables for the different years (Tables 3 and 4).

**Table 3.** DEA and DEA-BOOT results for efficiency.

| DMU No. | DMU Name | 2015 | | 2019 | |
|---|---|---|---|---|---|
| | | DEA | BOOT | DEA | BOOT |
| 1 | Barcelona | 1.000 | 0.704 | 1.000 | 0.723 |
| 2 | Tarragona | 0.023 | 0.020 | 0.335 | 0.306 |
| 3 | Castellón | 0.007 | 0.005 | 1.000 | 0.670 |
| 4 | Valencia | 0.345 | 0.293 | 0.413 | 0.358 |
| 5 | Baleares | 1.000 | 0.784 | 1.000 | 0.801 |
| 6 | Alicante | 0.127 | 0.115 | 0.119 | 0.105 |
| 7 | Cartagena | 0.332 | 0.275 | 0.654 | 0.574 |
| 8 | Almería | 1.000 | 0.643 | 0.105 | 0.096 |
| 9 | Motril | 0.072 | 0.068 | 0.093 | 0.085 |
| 10 | Málaga | 0.426 | 0.373 | 0.509 | 0.452 |
| 11 | Bahía de Cádiz | 1.000 | 0.681 | 1.000 | 0.841 |
| 12 | Huelva | 0.015 | 0.014 | 0.018 | 0.017 |

Source: own elaboration.

**Table 4.** Statistical results of DEA and DEA-BOOT analysis.

|  | 2015 | | 2019 | |
| --- | --- | --- | --- | --- |
|  | **DEA** | **BOOT** | **DEA** | **BOOT** |
| Arithmetic average | 0.445 | 0.331 | 0.520 | 0.419 |
| Geometric average | 0.178 | 0.145 | 0.315 | 0.268 |
| Standard deviation | 0.413 | 0.287 | 0.382 | 0.288 |
| Average deviation | 0.370 | 0.255 | 0.342 | 0.258 |
| Variance | 0.171 | 0.082 | 0.146 | 0.083 |

Source: own elaboration.

## 5. Discussion

The results obtained in Table 3 stand out, since out of the 12 port authorities studied, only 4 reached the DEA frontier value equal to 1.00, both for 2015 and 2019, with the same port authorities not matching in both years. However, the values obtained in the bootstrap analysis differed from approaching full efficiency, as the port authorities obtained very different values in their efficiency. It can be said that for bootstrap values higher than 0.7 (in this study), they were considered fully efficient.

The highest bootstrap values were reached by the port authority of the Balearic Islands (0.784) in 2015 and the port authority of the Bay of Cadiz (0.841) in 2019, reaching, in both cases, pure efficiency in the DEA analysis.

The lowest bootstrap values were attributed to the port authorities of Castellón (0.005) in 2015 and Huelva (0.017) in 2019. In the case of Castellón in 2015, this may be due to the few cruise passengers (366 cruise passengers) who stopped at its port compared to the rest of the ports, which reached well over a thousand, and in the case of Huelva, it is probably due to the decrease in the number of cruise passengers and the low number of cruise ship calls between its docks for the dimensions that these ships can receive over the course of a year.

Some of the efficiencies obtained values that have attracted the attention of researchers, such as the efficiency of Castellón, which improved compared to 2015, due to the increase in cruise passengers, as it is the port authority that grew the most in this study period, growing by more than 1.390%.

Almeria also stands out, but opposite of Castellón, it went from values that can be considered on the border of efficiency to values that are very far from being efficient. This is due, to a large extent, to the investment in the improvement of infrastructures for the berthing of cruise ships and to the sharp fall in the demand for cruise passengers in this port, with a drop in demand of more than 50% in the period under study.

Finally, the data obtained for the port authority of the Bay of Cadiz are curious, since in the DEA analysis, in both cases, it obtained the value of 1.000 (efficient), but in the boot-strap analysis, it went from a value on the efficiency frontier in 2015 to the highest value of all in 2019 with 0.841. This is mainly due to the investment in infrastructures, but without over-dimensioning the berth meters destined to this type of traffic, as other port authorities do, maintaining a fair and stable relationship between linear meters, cruise passengers, and cruise ships. Therefore, it is transmitted in obtaining good efficiency figures in this area of cruises.

Barcelona and the Balearic Islands, as expected, performed well, as they are the preferred destinations for cruise ships on many Western Mediterranean routes. They are also port authorities that host, and can host, a large number of large cruise ship calls throughout the year (including the world's largest, "Symphony of the Seas"), which also means a large number of cruise passengers disembarking (call or departure/finish) through them. This means that they are the only port authorities in the research to exceed 1.5 million cruise passengers.

*Case of Málaga*

Malaga, a port that is gradually consolidating its name as one of the ports to be taken into account in terms of cruise ships, obtained results that may seem significant, but which are far from reality. The bootstrap efficiencies obtained by the port authority of Malaga for the years 2015 and 2019 were 0.373 and 0.452, respectively. This, to a large extent, is the result of the correct work that the port has been carrying out in attracting this type of traffic in recent years, with heavy investments in the creation and modification of infrastructures to attract large cruise ships and a large number of cruise passengers to its port. This is a sign of the progressive evolution of the port in its investment in the future of being a large port of call for cruise ships in the Western Mediterranean as well as on the routes towards the Atlantic.

The port of Malaga has evolved a great deal in recent decades (see Table 5), going from a 450-meter berth in 2005 to tripling that size in 2008 with the creation of cruise pier B, with modern infrastructures and a renovated maritime station. Subsequently, in 2011, cruise terminal A was inaugurated, with more than 1600 m of berth dedicated exclusively to cruise traffic. This decision was taken in order to promote the port of Malaga as a base port for cruise ships, with large investments in infrastructures for this type of traffic and with large breakwaters, which help to ensure a good stay in the port.

**Table 5.** Evolution of the infrastructure destined for cruise traffic, as well as number of cruise ships and cruise passengers in Malaga.

| YEAR | Cruisers (Pax) | Cruises (Nº) | Cruises Docks (Meters) |
|------|---------------|--------------|------------------------|
| 2000 | 145,395 | 231 | 450.00 |
| 2005 | 400,103 | 213 | 450.00 |
| 2008 | 352,993 | 271 | 1175.00 |
| 2010 | 659,123 | 322 | 1595.00 |
| 2015 | 418,503 | 238 | 2045.00 |
| 2019 | 476,973 | 288 | 2045.00 |

Source: own elaboration.

In 2012, Pier 2, or the "Palmeral de las Sorpresas" pier, was inaugurated, which helped the port to accommodate certain cruise traffic, providing the port with more than 2000 m of quayside to berth and attract cruise ships.

The main problem or competitor is the Bay of Cadiz, which has also invested in being able to attract cruise traffic to its port, and that is why Malaga may not be able to consolidate itself 100% as a very attractive port for cruises.

Therefore, even if the efficiency obtained is low at the beginning, it can be said that the infrastructure is still "new", until the market is consolidated, and Malaga is able to establish itself as a major cruise port. For the moment, Malaga's work towards becoming a base port continues in a favorable manner, as it is currently a base port and is managing to sign new contracts that will consolidate it in the market as a major port for this type of traffic.

## 6. Conclusions

With the analysis of the DEA-bootstrapping model, the objective proposed with the methodology developed was achieved, and the desired results were extracted, obtaining satisfactory conclusions.

The efficiencies obtained show a diversified port system, not centered on cruise traffic in all its ports, which favors the dispersion of cruise ships along the entire coastline and the number of cruise passengers choosing their ports of origin.

The relevant data show that Barcelona and the Balearic Islands are very well-established as far as cruises are concerned, as their infrastructures, in relation to the traffic they attract and/or receive, are very efficient. They are ports that have been consolidated for years,

making them obligatory stopovers on the different Mediterranean routes, which makes them world references among the best.

There is also an improvement in certain efficiencies, however small, which makes ports want to attract this type of traffic to their ports. They represent income for the port and a tourist attraction for the city, so they invest in infrastructures so that certain cruise ships can dock, without investing to attract the new large mega-cruises.

Due to the improvement of infrastructures, the Spanish port system improves year after year in the number of cruise passengers who choose its ports to disembark and visit the different Spanish cities.

The Bay of Cadiz stands out for being more efficient in 2019, due to the improvements in port infrastructures for cruise traffic, being aware of the traffic it receives, which remains constant at around 400,000–500,000 cruise passengers, and being committed to being an important port of call.

Finally, in the case of Malaga, its efficiency is striking, and it is a sign that they are doing things well, due to the port's commitment to being a major base port in the Southern Mediterranean. For this reason, the large investments made in the last decade are bearing fruit, as they are gradually generating a greater demand from shipping companies who choose Malaga to start their routes, or even large cruise ships to call at the port. This gives Malaga an advantage over other possible competitors, as it has invested heavily in the last decade, and is fully capable of welcoming future generations of cruise ships that may arrive and of meeting the demand that the cruise sector may require, as its infrastructures are modern; in short, can adapt to the upcoming market, something for which other ports need large investments, or refuse contracts due to infrastructures deficiencies.

For future research, the objectives that can be taken into account are the types of ships that call at the main ports of the port system, as well as the economic impact that the consolidated ports in our port system have with regard to the cruise market. Finally, the analysis of the cruise market in the Mediterranean, with the appearance of restrictions such as ECAs or emission regulations, as well as changing trends in sustainability and emissions from ships and in ports, should also be taken into account for future lines of research.

**Author Contributions:** Conceptualization, all the authors; methodology, J.I.P.S. and A.C.O.; validation, all the authors; resources, J.I.P.S.; writing—original draft preparation, J.I.P.S., A.C.O. and D.D.G.; writing—review and editing, J.I.P.S. and A.C.O.; supervision, D.D.G. and F.d.A.D.M.L. All authors have read and agreed to the published version of the manuscript.

**Funding:** This research received no external funding.

**Institutional Review Board Statement:** The study did not require ethical approval.

**Informed Consent Statement:** Not applicable.

**Data Availability Statement:** The study did not report any data.

**Conflicts of Interest:** The authors declare no conflict of interest.

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
