# Peer review of "Employing DEA for Assessment of Cruise Market: A Case Study in Malaga—Spanish Port"

_jmse, doi:10.3390/jmse10121805_

Round 1

Reviewer 1 Report

This paper analysis interesting topic. The first of all language editing would be very beneficial.  Introduction section (state of the art) is written in confused way, so table containing important previous works, topics and results would be useful. Outcome of that table should be the proposal to survey this paper topic. Research design section/Methodology is not clearly described and it is not clear if there is certain novelty, so authors should rewrite this section and give more explanations. Also, more attention should be given to figures and formulas. The title of section where results are presented and analyzed should be divided to Results and Discussion. Also, limitations of this research and possible future research avenues in the conclusion section are missing, so should be added.

Author Response

Thank you very much for your comments.
Thanks to them we have been able to improve our paper.
We have corrected aspects such as: including a summary table with the main research in the cruise market; we have defined the methodology making clear the contribution of the research; we have eliminated some tables that did not add any value to the paper; and in the conclusions we have added the future lines to follow.
Thank you again for your recommendations. 

Reviewer 2 Report

Please read the attached file. Thank you.

Author Response

Thank you reviewer for the changes made.
In conjunction with your comments, those of the other reviewer of the paper have also been made.
All the points mentioned in your review list have been corrected, except for the decimals of the analysis: instead of two decimals, we have decided to leave it at 3 decimals (instead of the original 5 decimals). 
We proceed to answer your questions:
1) Because the great "boom"/growth of cruise traffic has been happening in the first decade of the 21st century and therefore we have come to analyse the previous sequence of adaptation and progress up to 2019, with qualitative leaps; since in different years similar data, the authors want to capture the progress in terms of cruises in the port of Malaga.
2) DEA has been chosen because the Bootstrap technique provides estimates of the statistical error imposing few restrictions on the random variables analysed and establishing itself as a general procedure, regardless of the statistic considered.

3) Thanks to your comment, some references have been added regarding the MDCM, as well as other research you have done regarding efficiencies; but they have not been taken into account as you explain in your comments.

4) With the DEA model used, the results have not been compared, this research comes to explain a current trend in the cruise market in the Spanish port system, and the reader of this research is made to see the comparison and improvement in efficiency of the ports whose cruise traffic is most relevant.

5) It have been added in the conclusions, which was also a request of the other reviewer.

Round 2

Reviewer 1 Report

The paper is significantly improved and could be accepted.

Reviewer 2 Report

Dear Editor and Authors: 

Thank you for your responses. 

The authors have corrected and answered all my comments and questions. It sounds good now. The reviewer suggests that it should be accepted for publication. 

Thank you. 

Sincerely yours, 

The reviewer.